# StreamingT2V: Consistent, Dynamic, and Extendable Long Video Generation from Text

## Abstract

Text-to-video diffusion models enable the generation of high-quality videos that follow text instructions, simplifying the process of producing diverse and individual content. Current methods excel in generating short videos (up to 16s), but produce hard-cuts when naively extended to long video synthesis. To overcome these limitations, we present *StreamingT2V*, an autoregressive method that generates long videos of **up to 2 minutes or longer** with seamless transitions. The key components are: (i) a short-term memory block called conditional attention module (CAM), which conditions the current generation on the features extracted from the preceding chunk via an attentional mechanism, leading to consistent chunk transitions, (ii) a long-term memory block called appearance preservation module (APM), which extracts high-level scene and object features from the first video chunk to prevent the model from forgetting the initial scene, and (iii) a randomized blending approach that allows for the autoregressive application of a video enhancer on videos of indefinite length, ensuring consistency across chunks. Experiments show that StreamingT2V produces high motion amount, while competing methods suffer from video stagnation when applied naively in an autoregressive fashion. Thus, we propose with StreamingT2V a high-quality seamless text-to-long video generator, surpassing competitors in both consistency and motion.

## 1 Introduction

The emergence of diffusion models Ho et al. (2020); Song et al. (2020); Rombach et al. (2022); Ramesh et al. (2022) has sparked significant interest in text-guided image synthesis and manipulation. Building on the success in image generation, they have been extended to text-guided video generation Ho et al. (2022b); Wang et al. (2023b); Blattmann et al. (2023b); Chen et al. (2023b); Singer et al. (2022); Girdhar et al. (2023); Blattmann et al. (2023a); Guo et al. (2023b); Li et al. (2023); Zhang et al. (2023a); Guo et al. (2023a); Khachatryan et al. (2023); Villegas et al. (2022).

Despite the impressive generation quality and text alignment, the majority of existing approaches such as Ho et al. (2022b); Wang et al. (2023b); Blattmann et al. (2023b;a); Zhang et al. (2023a); Zheng et al. (2024); PKU-Yuan-Lab & Tuzhan-AI (2024) are mostly focused on generating short frame sequences (typically of $16$, $24$, or recently $384$ frame-length). However, short videos generators are limited in real-world use-cases such as ad making, storytelling, etc.

The naïve approach of training video generators on long videos (*e.g.* $\geq 1200$ frames) is usually impractical. Even for generating short sequences, it typically requires expensive training (*e.g.* $260K$ steps and $4.5K$ batch size in order to generate 16 frames Wang et al. (2023b)).

Some approaches Oh et al. (2023); Blattmann et al. (2023b); Ho et al. (2022b); Zheng et al. (2024) thus extend the baselines to autoregressively generate short video chunks conditioned on the last frame(s) of the preceding chunk. Yet, simply concatenating the noisy latents of a video chunk with the last frame(s) of the preceding chunk leads to poor conditioning with inconsistent scene transitions (see Sec. D). Some works Blattmann et al. (2023a); Zhang et al. (2023c); Wang et al. (2024); Dai et al. (2023); Xing et al. (2023) integrate also CLIP Radford et al. (2021) image embeddings of the last frame of the preceding chunk, which slightly improves consistency. However, they are still prone to inconsistencies across chunks (see Fig. 12) due to the CLIP image encoder losing crucial information necessary for accurately reconstructing the conditional frames. The concurrent

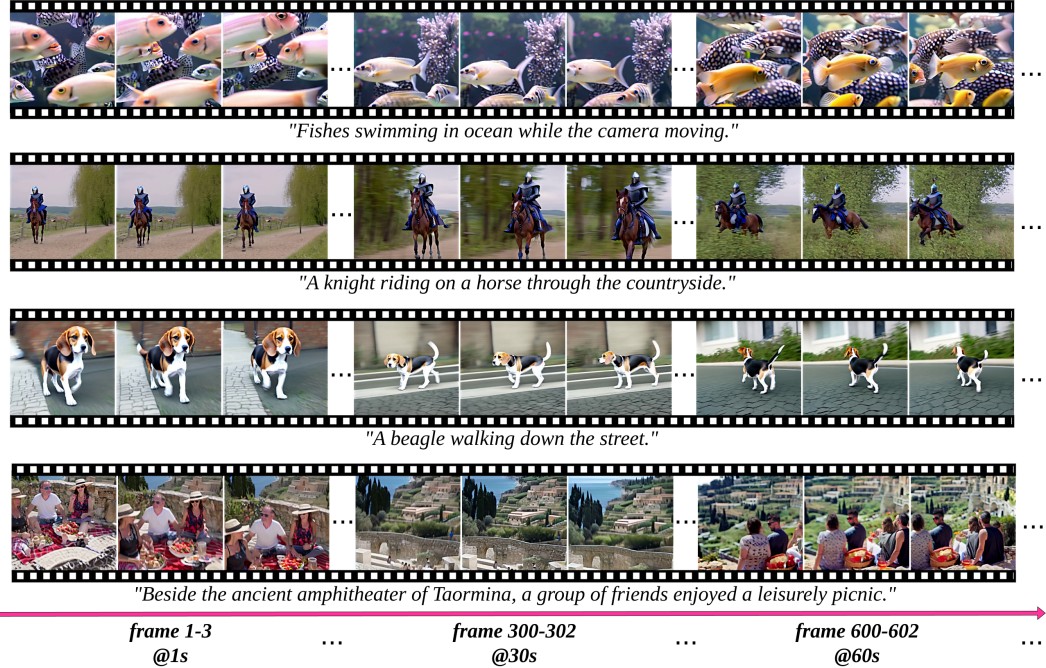

"Fishes swimming in ocean while the camera moving."

"A knight riding on a horse through the countryside."

"A beagle walking down the street."

"Beside the ancient amphitheater of Taormina, a group of friends enjoyed a leisurely picnic."

frame 1-3    …    frame 300-302    …    frame 600-602    …
@1s                @30s                 @60s

Figure 1: **StreamingT2V** is an advanced autoregressive technique to create long videos featuring rich motion dynamics, ensuring temporal consistency, alignment with descriptive text, high frame-level image quality, and no stagnation. Demonstrations include videos up to **1200 frames, spanning 2 minutes**, which can be extended further. The effectiveness of StreamingT2V is not limited by the Text2Video model used, indicating potential for even higher-quality with improved base models.

work SparseCtrl Guo et al. (2023a) utilizes a more sophisticated conditioning mechanism by sparse encoder. To match the size of the inputs, its architecture requires to concatenate additional zero-filled frames to the conditioning frames before being plugged into sparse encoder. However, this inconsistency in the input leads to inconsistencies in the output (see Sec. 5.3).

Our experiments (see Sec. 5.3) reveal that in fact all-image-to-video methods we assessed eventually result in video stagnation or strong quality degradation when applied autoregressively by conditioning on the last frame of the preceding chunk.

To overcome the weaknesses and limitations of current works, we propose *StreamingT2V*, an autoregressive text-to-video method equipped with long/short-term memory blocks that generates long videos without temporal inconsistencies.

To this end, we propose the *Conditional Attention Module (CAM)* which, due to its attentional nature, effectively borrows the content information from the previous frames to generate new ones, while not restricting their motion by the previous structures/shapes. Thanks to CAM, our results are smooth and with artifact-free video chunk transitions.

Current methods not only exhibit temporal inconsistencies and video stagnation, but also experience alterations in object appearance/characteristics (see *e.g.* SEINE Chen et al. (2023b) in Fig. 9) and a decline in video quality over time (see *e.g.* SVD Blattmann et al. (2023a) in Fig. 5). This occurs as only the last frame(s) of the preceding chunk are considered, thus overlooking long-term dependencies in the autoregressive process. To address this issue we design an *Appearance Preservation Module (APM)* that extracts object and global scene details from an initial image, to condition the video generation with that information, ensuring consistency in object and scene features throughout the autoregressive process.

To further enhance the quality and resolution of our long video generation, we adapt a video enhancement model for autoregressive generation. To this end, we select a high-resolution text-to-video model and apply the SDEdit Meng et al. (2022) approach to enhance consecutive 24-frame

chunks (overlapping with 8 frames) of our video. To make the chunk enhancement transitions smooth, we design a ***randomized blending*** approach for seamless merging of overlapping chunks.

Experiments show that StreamingT2V successfully generates long and temporal consistent videos from text without video stagnation. To summarize, our contributions are three-fold:

- We introduce ***StreamingT2V***, an autoregressive approach for seamless synthesis of extended video content using short and long-term dependencies.
- Our ***Conditional Attention Module (CAM)*** and ***Appearance Preservation Module (APM)*** ensure the natural continuity of the global scene and object characteristics of generated videos.
- We seamlessly enhance generated long videos by introducing our ***randomized blending approach*** of consecutive overlapping chunks.

## 2 RELATED WORK

**Text-Guided Video Diffusion Models.** Generating videos from text instructions using Diffusion Models Ho et al. (2020); Sohl-Dickstein et al. (2015) is a newly established and actively researched field introduced by Video Diffusion Models (VDM) Ho et al. (2022b). The method can generate only low-resolution videos (up to 128x128) with a maximum of 16 frames (without autoregression), imposing significant limitations, while requiring massive training resources. Several methods thus employ video enhancement in the form of spatial/temporal upsampling Ho et al. (2022a); Singer et al. (2022); Ho et al. (2022b); Blattmann et al. (2023b), using cascades with up to 7 enhancer modules Ho et al. (2022a). While this leads to high-resolution and long videos, the generated content is still limited by the content depicted in the key frames.

Towards generating longer videos (*i.e.* more keyframes), Text-To-Video-Zero (T2V0) Khachatryan et al. (2023) and ART-V Weng et al. (2023) utilize a text-to-image diffusion model. Therefore, they can generate only simple motions. T2V0 conditions on its first frame via cross-frame attention and ART-V on an anchor frame. Due to the lack of global reasoning, it leads to unnatural or repetitive motions. MTVG Oh et al. (2023) transforms a text-to-video model into an autoregressive method through a training-free approach. As it uses strong consistency priors within and between chunks, it results in very low motion amount, and mostly near-static background. FreeNoise Qiu et al. (2024) samples a small set of noise vectors, re-uses them for the generation of all frames, while temporal attention is performed on local windows. As temporal attention is invariant to such frame shuffling, it leads to high similarity between frames, almost always static global motion and near-constant videos. Gen-L Wang et al. (2023a) generates overlapping short videos and combines them via temporal co-denoising, which can lead to quality degradation with video stagnation. Recent transformed-based diffusion models Zheng et al. (2024); PKU-Yuan-Lab & Tuzhan-AI (2024) operate in the latent space of a 3D autoencoder, enabling the generation of up to 384 frames. Despite extensive training, these models produce videos with limited motion, often resulting in near-constant videos.

**Image-Guided Video Diffusion Models as Long Video Generators.** Several works condition the video generation by a driving image or video Xing et al. (2023); Chen et al. (2023b); Blattmann et al. (2023a); Guo et al. (2023a); Esser et al. (2023); Zhang et al. (2023c); Chen et al. (2023a); Zeng et al. (2023); Dai et al. (2023); Wang et al. (2024); Long et al. (2024); Ren et al. (2024). They can thus be turned into an autoregressive method by conditioning on the frame(s) of the preceding chunk.

VideoDrafter Long et al. (2024) uses a text-to-image diffusion model to obtain an anchor frame. A video diffusion model is conditioned on the driving anchor to generate independently multiple videos that share the same high-level context. However, this leads to drastic scene cuts as no consistency among video chunks is enforced. StoryDiffusion Zhou et al. (2024) conditions on video frames that have been linearly propagated from key frames, which leads to severe quality degradation. Several works Chen et al. (2023b); Zeng et al. (2023); Dai et al. (2023) concatenate the (encoded) conditionings (*e.g.* input frame(s)) with an additional mask (indicating the provided frame(s)) to the input of the video diffusion model.

In addition to concatenating the conditioning to the input of the diffusion model, several works Blattmann et al. (2023a); Zhang et al. (2023c); Wang et al. (2024) replace the text embeddings in

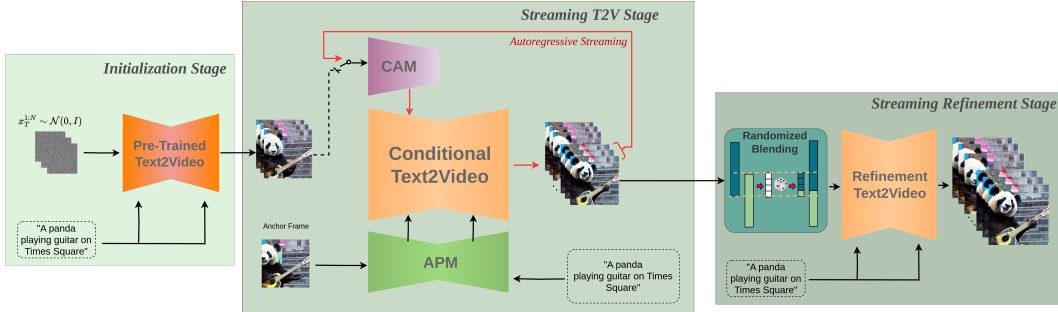

Figure 2: The overall pipeline of StreamingT2V involves three stages: (i) *Initialization Stage*: The first 16-frame chunk is synthesized by an off-the-shelf text-to-video model. (ii) *Streaming T2V Stage*: New content for subsequent frames is autoregressively generated. (iii) *Streaming Refinement Stage*: The long video (*e.g.* 240, 1200 frames or more) is autoregressively enhanced using a high-resolution text-to-short-video model with a randomized blending approach.

the cross-attentions of the diffusion model by CLIP Radford et al. (2021) image embeddings of the conditional frames. However, according to our experiments, their applicability for long video generation is limited. SVD Blattmann et al. (2023a) shows severe quality degradation over time (see Fig. 5), and both, I2VGen-XL Zhang et al. (2023c) and SVD Blattmann et al. (2023a) generate often inconsistencies between chunks, still indicating that the conditioning mechanism is too weak.

Some works Xing et al. (2023); Chen et al. (2023a) such as DynamiCrafter-XL Xing et al. (2023) thus add to each text cross-attention an image cross-attention, which leads to better quality, but still to frequent inconsistencies between chunks.

The concurrent work SparseCtrl Guo et al. (2023a) adds a ControlNet Zhang et al. (2023b)-like branch to the model, taking the conditional frames and a frame-mask as input. It requires by design to append additional frames consisting of black pixels to the conditional frames. This inconsistency is difficult to compensate for the model, leading to frequent and severe scene cuts between frames.

Overall, only a small number of keyframes can currently be generated at once with high quality. While in-between frames can be interpolated, it does not lead to new content. Also, while image-to-video methods can be used autoregressively, their used conditional mechanisms lead either to inconsistencies, or the method suffers from video stagnation. We conclude that existing works are not suitable for high-quality and consistent long video generation without video stagnation.

## 3 PRELIMINARIES

**Diffusion Models.** Our text-to-video model, which we term *StreamingT2V*, is a diffusion model that operates in the latent space of the VQ-GAN Esser et al. (2021); Van Den Oord et al. (2017) autoencoder $\mathcal{D}(\mathcal{E}(\cdot))$, where $\mathcal{E}$ and $\mathcal{D}$ are the corresponding encoder and decoder, respectively. Given a video $\mathcal{V} \in \mathbb{R}^{F \times H \times W \times 3}$, composed of $F$ frames with spatial resolution $H \times W$, its *latent code* $x_0 \in \mathbb{R}^{F \times h \times w \times c}$ is obtained through frame-wise application of the encoder. More precisely, by identifying each tensor $x \in \mathbb{R}^{F \times \hat{h} \times \hat{w} \times \hat{c}}$ as a sequence $(x^f)_{f=1}^F$ with $x^f \in \mathbb{R}^{\hat{h} \times \hat{w} \times \hat{c}}$, we obtain the latent code via $x_0^f := \mathcal{E}(\mathcal{V}^f)$, for all $f = 1, \ldots, F$. The diffusion forward process gradually adds Gaussian noise $\epsilon \sim \mathcal{N}(0, I)$ to the signal $x_0$:

$$q(x_t|x_{t-1}) = \mathcal{N}(x_t; \sqrt{1 - \beta_t} x_{t-1}, \beta_t I), \ t = 1, \ldots, T \tag{1}$$

where $q(x_t|x_{t-1})$ is the conditional density of $x_t$ given $x_{t-1}$, and $\{\beta_t\}_{t=1}^T$ are hyperparameters. A high value for $T$ is chosen such that the forward process completely destroys the initial signal $x_0$ resulting in $x_T \sim \mathcal{N}(0, I)$. The goal of a diffusion model is then to learn a backward process

$$p_\theta(x_{t-1}|x_t) = \mathcal{N}(x_{t-1}; \mu_\theta(x_t, t), \Sigma_\theta(x_t, t)) \tag{2}$$

for $t = T, \ldots, 1$ (see DDPM Ho et al. (2020)), which allows to generate a valid signal $x_0$ from standard Gaussian noise $x_T$. Once $x_0$ is obtained from $x_T$, the generated video is obtained by

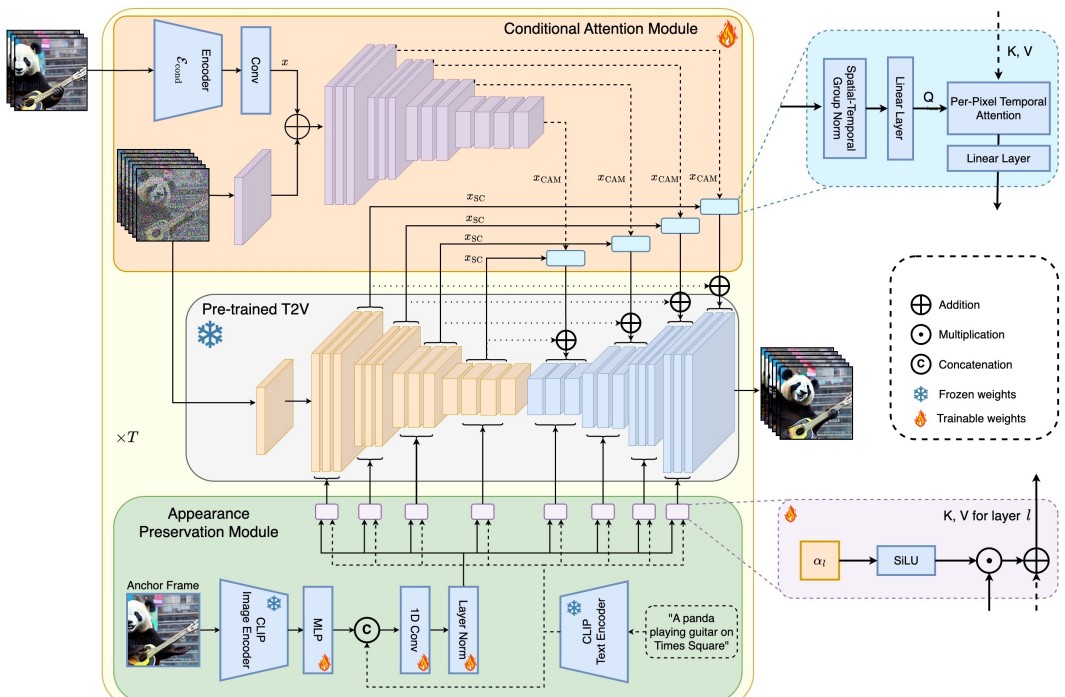

Figure 3: Method overview: StreamingT2V enhances a video diffusion model (VDM) with the conditional attention module (CAM) for short-term memory, and with the appearance preservation module (APM) for long-term memory. CAM conditions a VDM on the preceding chunk using a frame encoder $\mathcal{E}_{\text{cond}}$. CAM's attentional mechanism enables smooth transitions between chunks *and* high motion. APM extracts high-level image features from an anchor frame and injects them into the text cross-attentions of the VDM, preserving object/scene features during the autoregression.

applying the decoder frame-wise: $\widetilde{\mathbf{V}}^f := \mathcal{D}(x_0^f)$, for all $f = 1, \ldots, F$. Yet, instead of learning a predictor for mean and variance in Eq. 2, we learn a model $\epsilon_\theta(x_t, t)$ to predict the Gaussian noise $\epsilon$ that was used to form $x_t$ from input signal $x_0$ (a common reparametrization Ho et al. (2020)).

For text-guided video generation, we use a neural network with learnable weights $\theta$ as noise predictor $\epsilon_\theta(x_t, t, \tau)$ that is conditioned on the textual prompt $\tau$. We train it on the denoising task:

$$\min_\theta \mathbb{E}_{t,(x_0,\tau)\sim p_{data},\epsilon\sim\mathcal{N}(0,I)}||\epsilon - \epsilon_\theta(x_t, t, \tau)||_2^2, \tag{3}$$

using the data distribution $p_{data}$. To simplify notation, we will denote by $x_t^{r:s} := (x_t^j)_{j=r}^s$ the latent sequence of $x_t$ from frame $r$ to frame $s$, for all $r, t, s \in \mathbb{N}$.

**Text-To-Video Models.** Text-to-video models Girdhar et al. (2023); Singer et al. (2022); Wang et al. (2023b); Ho et al. (2022a); Blattmann et al. (2023b) typically expand pre-trained text-to-image models Rombach et al. (2022); Ramesh et al. (2022) by adding new layers that operate on the temporal axis. Modelscope (MS) Wang et al. (2023b) follows this approach by extending the UNet-like Ronneberger et al. (2015) architecture of Stable Diffusion Rombach et al. (2022) with temporal convolutional and attentional layers. It was trained in a large-scale setup to generate videos with 3 FPS@256x256 and 16 frames. The quadratic growth in memory and compute of the temporal attention layers (as used in recent text-to-video models) together with very high training costs limits current text-to-video models in generating long sequences. In this paper, we demonstrate our StreamingT2V method by taking MS as a basis and turn it into an autoregressive model suitable for long video generation with high motion dynamics and consistency.

## 4 METHOD

In this section, we introduce our method for high-resolution text-to-long video generation. We first generate $256 \times 256$ resolution long videos (240 frames, or 1200 frames), then enhance them to higher resolution ($720 \times 720$). The overview of the whole pipeline is provided in Fig. 2. The long video generation process comprises three stages: the ***Initialization Stage***, where the first 16-frame chunk is synthesized by a pre-trained text-to-video model (*e.g.* Modelscope Wang et al. (2023b)), the ***Streaming T2V Stage*** where new content for subsequent frames is generated autoregressively. To ensure seamless transitions between chunks, we introduce (see Fig. 3) our conditional attention module (CAM), which utilizes short-term information from the last $F_{cond} = 8$ frames and our appearance preservation module (APM), which extracts long-term information from an anchor frame to maintain object appearance and scene details during the autoregressive process. After generating a long video (*e.g.* 240, 1200 frames or more), the ***Streaming Refinement Stage*** enhances the video using a high-resolution text-to-short-video model (*e.g.* MS-Vid2Vid-XL Zhang et al. (2023c)) autoregressively with our randomized blending approach for seamless chunk processing. This step does not require additional training, making our approach cost-effective.

### 4.1 CONDITIONAL ATTENTION MODULE

For training a conditional network in our Streaming T2V stage, we leverage the capabilities of a pre-trained text-to-video model (*e.g.* Modelscope Wang et al. (2023b)) as a prior for autoregressive long video generation. Subsequently, we will denote this pre-trained text-to-(short)video model as ***Video-LDM***. To condition Video-LDM autoregressively with short-term information from the preceding chunk (see Fig. 2, mid), we introduce the ***Conditional Attention Module (CAM)***. CAM consists of a feature extractor and a feature injector into the Video-LDM UNet, inspired by ControlNet Zhang et al. (2023b). The feature extractor utilizes a frame-wise image encoder $\mathcal{E}_{\text{cond}}$, followed by the same encoder layers that the Video-LDM UNet uses up to its middle layer (initialized with the UNet's weights). For the feature injection, we let each long-range skip connection in the UNet *attend* to corresponding features generated by CAM via cross-attention.

Let $x$ denote the output of $\mathcal{E}_{\text{cond}}$ after zero-convolution. We use addition to fuse $x$ with the output of the first temporal transformer block of CAM. For the injection of CAM's features into the Video-LDM Unet, we consider the UNet's skip-connection features $x_{\text{SC}} \in \mathbb{R}^{b \times F \times h \times w \times c}$ (see Fig. 3) with batch size $b$. We apply spatio-temporal group norm, and a linear projection $P_{\text{in}}$ on $x_{\text{SC}}$. Let $x'_{\text{SC}} \in \mathbb{R}^{(b \cdot w \cdot h) \times F \times c}$ be the resulting tensor after reshaping. We condition $x'_{\text{SC}}$ on the corresponding CAM feature $x_{\text{CAM}} \in \mathbb{R}^{(b \cdot w \cdot h) \times F_{\text{cond}} \times c}$ (see Fig. 3), where $F_{\text{cond}}$ is the number of conditioning frames, via temporal multi-head attention (T-MHA) Vaswani et al. (2017), *i.e.* independently for each spatial position (and batch). Using learnable linear maps $P_Q, P_K, P_V$, for queries, keys, and values, we apply T-MHA using keys and values from $x_{\text{CAM}}$ and queries from $x'_{\text{SC}}$, *i.e.*

$$x''_{\text{SC}} = \text{T-MHA}\big(Q = P_Q(x'_{\text{SC}}), K = P_K(x_{\text{CAM}}), V = P_V(x_{\text{CAM}})\big). \qquad (4)$$

Finally, we use a linear projection $P_{out}$. Using a suitable reshaping operation $R$, the output of CAM is added to the skip connection (as in ControlNet Zhang et al. (2023b)):

$$x'''_{\text{SC}} = x_{\text{SC}} + R(P_{\text{out}}(x''_{\text{SC}})), \qquad (5)$$

so that $x'''_{\text{SC}}$ is used in the decoder layers of the UNet. We use zero-initialized projection $P_{\text{out}}$, so that CAM initially does not affect the base model's output, which improves convergence.

The design of CAM enables conditioning the $F$ frames of the base model on the $F_{\text{cond}}$ frames of the preceding chunk. In contrast, sparse encoder Guo et al. (2023a) employs convolution for feature injection, thus needs additional $F - F_{\text{cond}}$ zero-valued frames (and a mask) as input, in order to add the output to the $F$ frames of the base model. These inconsistencies in the input lead to severe inconsistencies in the output (see Sec. D.1 and Sec. 5.3).

### 4.2 APPEARANCE PRESERVATION MODULE

Autoregressive video generators typically suffer from forgetting initial object and scene features, leading to severe appearance changes. To tackle this issue, we incorporate long-term memory by

leveraging the information contained in a fixed anchor frame of the very first chunk using our proposed **Appearance Preservation Module (APM)**. This helps to maintain scene and object features across video chunk generations (see Fig. 13).

To enable APM to balance guidance from the anchor frame and the text instructions, we propose (see Fig. 3): (i) We combine the CLIP Radford et al. (2021) image token of the anchor frame with the CLIP text tokens from the textual instruction by expanding the clip image token to $k = 16$ tokens using an MLP layer, concatenating the text and image encodings at the token dimension, and utilizing a projection block, leading to $x_{\text{mixed}} \in \mathbb{R}^{b \times 77 \times 1024}$; (ii) For each cross-attention layer $l$, we introduce a weight $\alpha_l \in \mathbb{R}$ (initialized as 0) to perform cross-attention using keys and values derived from a weighted sum $x_{\text{mixed}}$, and the usual CLIP text encoding of the text instructions $x_{\text{text}}$:

$$x_{\text{cross}} = \text{SiLU}(\alpha_l) x_{\text{mixed}} + x_{\text{text}}. \tag{6}$$

The experiments in Sec. D.2 show that the light-weight APM module helps to keep scene and identity features across the autoregressive process (see Fig. 13).

### 4.3 Auto-regressive Video Enhancement

To further enhance the quality and resolution of our text-to-video results, we use a high-resolution ($1280 \times 720$) text-to-(short)video model (Refiner Video-LDM, see Fig. 2), *e.g.* MS-Vid2Vid-XL Wang et al. (2024); Zhang et al. (2023c), to autoregressively improve 24-frame video chunks. To this end, we add noise to each video chunk and then denoise it using Refiner Video-LDM (SDEdit approach Meng et al. (2022)). Specifically, we upscale each low-resolution 24-frame video chunk to $720 \times 720$ using bilinear interpolation Amidror (2002), zero-pad to $1280 \times 720$, encode the frames with the image encoder $\mathcal{E}$ to get a latent code $x_0$, apply $T' < T$ forward diffusion steps (see Eq. 1) so that $x_{T'}$ still contains signal information, and denoise it with Refiner Video-LDM.

Naively enhancing each chunk independently leads to inconsistent transitions (see Fig. 4 (a)). To overcome this shortcoming, we introduce shared noise and a randomized blending technique. We divide a low-resolution long video into $m$ chunks $\mathcal{V}_1, \ldots, \mathcal{V}_m$ of $F = 24$ frames, each with an $\mathcal{O} = 8$ frames overlap between consecutive chunks. For each denoising step, we must sample noise (compare Eq. 2). We combine that noise with the noise already sampled for the overlapping frames of the preceding chunk to form **shared noise**. Specifically, for chunk $\mathcal{V}_i$, $i = 1$, we sample noise $\epsilon_1 \sim \mathcal{N}(0, I)$ with $\epsilon_1 \in \mathbb{R}^{F \times h \times w \times c}$. For $i > 1$, we sample noise $\hat{\epsilon}_i \sim \mathcal{N}(0, I)$ with $\hat{\epsilon}_i \in \mathbb{R}^{(F-\mathcal{O}) \times h \times w \times c}$ and concatenate it with $\epsilon_{i-1}^{(F-\mathcal{O}):F}$ (already sampled for the preceding chunk) along the frame dimension to obtain $\epsilon_i$ *i.e.*:

$$\epsilon_i := \text{concat}([\epsilon_{i-1}^{(F-\mathcal{O}):F}, \hat{\epsilon}_i], \dim = 0). \tag{7}$$

At diffusion step $t$ (starting from $T'$), we perform one denoising step using $\epsilon_i$ and obtain for chunk $\mathcal{V}_i$ the latent code $x_{t-1}(i)$. Despite these efforts, transition misalignment persists (see Fig. 4 (b)).

To significantly improve consistency, we introduce *randomized blending*. Consider the latent codes $x_L := x_{t-1}(i-1)$ and $x_R := x_{t-1}(i)$ of two consecutive chunks $\mathcal{V}_{i-1}, \mathcal{V}_i$ at denoising step $t-1$. The latent code $x_L$ of chunk $\mathcal{V}_{i-1}$ possesses a smooth transition from its first frames to the overlapping frames, while the latent code $x_R$ possesses a smooth transition from the overlapping frames to the subsequent frames. Thus, we combine the two latent codes via concatenation at a randomly chosen overlap position, by randomly sampling a frame index $f_{\text{thr}}$ from $\{0, \ldots, \mathcal{O}\}$ according to which we merge the two latents $x_L$ and $x_R$:

$$x_{LR} := \text{concat}([x_L^{1:F-f_{\text{thr}}}, x_R^{f_{\text{thr}}+1:F}], \dim = 0). \tag{8}$$

Then, we update the latent code of the entire long video $x_{t-1}$ on the overlapping frames and perform the next denoising step. Accordingly, for a frame $f \in \{1, \ldots, \mathcal{O}\}$ of the overlap, the latent code of chunk $\mathcal{V}_{i-1}$ is used with probability $1 - \frac{f}{\mathcal{O}+1}$. This probabilistic mixture of latents in overlapping regions effectively diminishes inconsistencies between chunks (see Fig. 4(c)). The importance of randomized blending is further assessed in an ablation study in the appendix (see Sec. D).

## 5 Experiments

We elaborate on our implementation, and present qualitative and quantitative evaluations. The importance of our contributions are supported by ablation studies in the appendix (Sec. D).

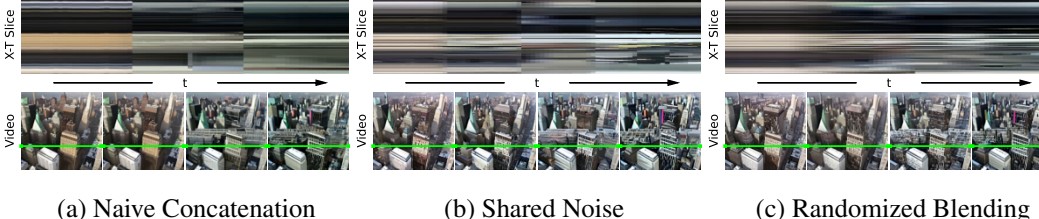

(a) Naive Concatenation       (b) Shared Noise       (c) Randomized Blending

Figure 4: Ablation study on our video enhancer improvements. The X-T slice visualization shows that randomized blending leads to smooth chunk transitions, while both baselines have clearly visible, severe inconsistencies between chunks.

## 5.1 IMPLEMENTATION DETAILS

We generate $F = 16$ frames, condition on $F_{\text{cond}} = 8$ frames, and display the video with 10 FPS. Training is conducted using a dataset collected from publicly available sources. We sample with 3FPS@256x256 16 frames (during CAM training) and 32 frames (during CAM+APM training).

**CAM training**: we freeze the weights of the pre-trained Video-LDM and train the new layers of CAM with batch size 8 and learning rate $5 \cdot 10^{-5}$ for 400K steps.
**CAM+APM training**: After the CAM training, we freeze the CLIP encoder and the temporal layers of the main branch, and train the remaining layers for 1K steps.

The image encoder $\mathcal{E}_{\text{cond}}$ used in CAM is composed of stacked 2D convolutions, layer norms and SiLU activations. For the video enhancer, we diffuse an input video using $T' = 600$ steps. Further training and implementation details are provided in the appendix (see Sec. E).

## 5.2 METRICS

For quantitative evaluation we employ metrics that measure temporal consistency, text-alignment, and per-frame quality of our method.

For temporal consistency, we introduce SCuts, which counts the number of detected scene cuts in a video using the AdaptiveDetector PyS with default parameters. In addition, we propose a new metric called **motion aware warp error (MAWE)**, which coherently assesses motion amount and warp error, and yields a low value when a video exhibits both consistency *and* a substantial amount of motion. To this end, we measure the motion amount using OFS (optical flow score), which computes for a video the mean of the squared magnitudes of all optical flow vectors between any two consecutive frames. Furthermore, for a video $\mathcal{V}$, we consider the mean warp error Lai et al. (2018) $W(\mathcal{V})$, which measures the average squared L2 pixel distance from a frame to its warped subsequent frame, excluding occluded regions. Finally, MAWE is defined as:

$$\text{MAWE}(\mathcal{V}) := \frac{W(\mathcal{V})}{\text{OFS}(\mathcal{V})}, \qquad (9)$$

which we found to be well-aligned with human perception. For the metrics involving optical flow, computations are conducted by resizing all videos to $720 \times 720$ resolution.

For video textual alignment, we employ the CLIP Radford et al. (2021) text image similarity score (CLIP), which is applied to all frames of a video. CLIP computes for a video sequence the cosine similarity from the CLIP text encoding to the CLIP image encodings.

For per-frame quality we incorporate the Aesthetic Score Schuhmann et al. (2022), which is computed on top of CLIP image embeddings of all frames of a video.

All metrics are computed per video first and then averaged over all videos, all videos are generated with 240 frames for quantitative analysis.

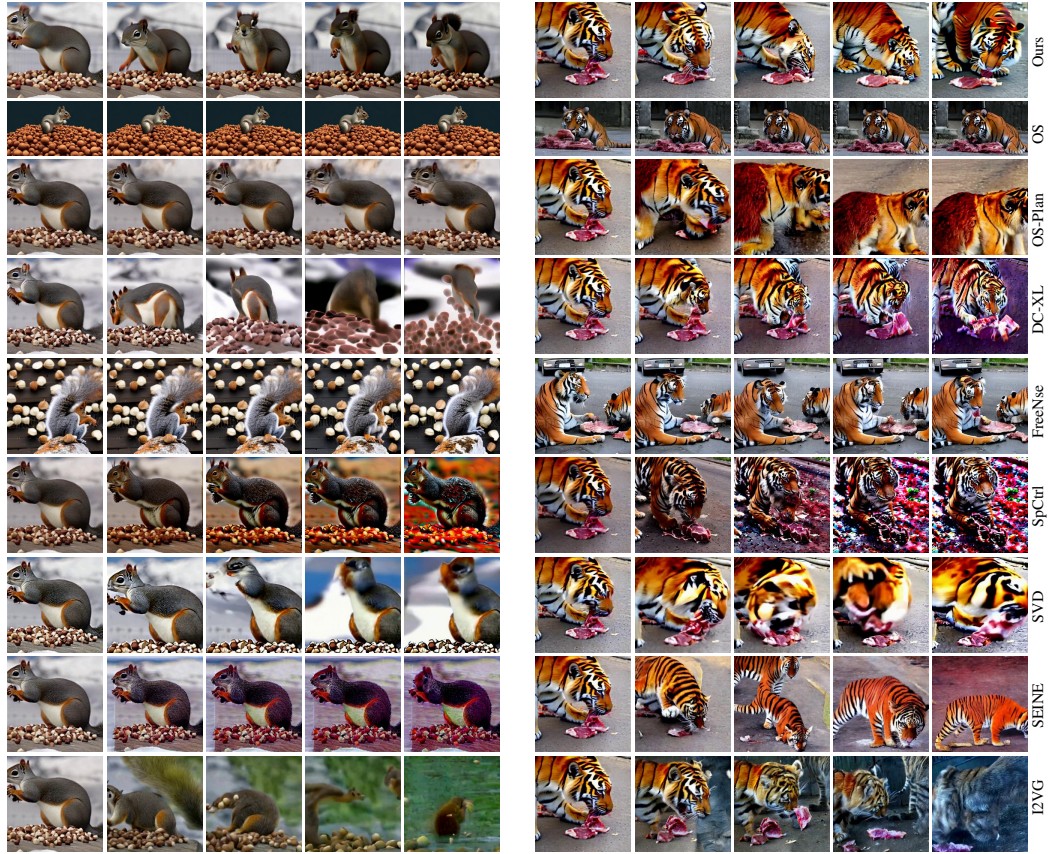

(a) A squirrel in Antarctica, on a pile of hazelnuts.

(b) A tiger eating raw meat on the street.

Figure 5: Visual comparisons of StreamingT2V with state-of-the-art methods on 240 frame-length, autoregressively generated videos. In contrast to other methods, StreamingT2V generates long videos without suffering from motion stagnation.

## 5.3 COMPARISON WITH BASELINES

**Benchmark.** To assess the effectiveness of StreamingT2V, we created a test set composed of 50 prompts covering different actions, objects and scenes (prompts are listed in Sec. F). We compare against recent methods that have code available, namely the image-to-video methods I2VGen-XL Zhang et al. (2023c), SVD Blattmann et al. (2023a), DynamiCrafter-XL Xing et al. (2023), OpenSoraPlan v1.2 PKU-Yuan-Lab & Tuzhan-AI (2024) and SEINE Chen et al. (2023b) used autoregressively, the video-to-video methods SparseControl Guo et al. (2023a), OpenSora v1.2 Zheng et al. (2024), and FreeNoise Qiu et al. (2024).

For all methods, we use their released model weights and hyperparameters. To have a fair comparison and insightful analysis on the performance of the methods for the autoregressive generation, and make the analysis independent on the employed initial frame generator, we use the same Video-LDM model to generate the first chunk consisting of 16 frames, given a text prompt and enhance it to 720x720 resolution using the same Refiner Video-LDM. Then, we generate the videos, while we start all autoregressive methods by conditioning on the last frame(s) of that chunk. For methods working on different spatial resolution, we apply zero padding to the initial frame(s). All evaluations are conducted on 240-frames video generations.

**Automatic Evaluation.** Our quantitative evaluation on the test set shows that StreamingT2V clearly performs best in terms of seamless chunk transitions and motion consistency (see Tab. 6). Our MAWE score significantly excels all competing methods (*e.g.* nearly 30% lower than the second

Table 6: Quantitative comparison to state-of-the-art open-source text-to-long-video generators. Best performing metrics are highlighted in red, second best in blue. Our method performs best in MAWE and CLIP score. Only in SCuts, StreamingT2V scores second best, as FreeNoise generates near-constant videos.

| Method | ↓MAWE | ↓SCuts | ↑CLIP |
| --- | --- | --- | --- |
| | More motion / less stagnation | Better consistency / less scene change | Better text alignment |
| SparseCtrl Guo et al. (2023a) | 6069.7 | 5.48 | 29.32 |
| I2VGenXL Zhang et al. (2023c) | 2846.4 | 0.4 | 27.28 |
| DynamiCrafterXL Xing et al. (2023) | 176.7 | 1.3 | 27.79 |
| SEINE Chen et al. (2023b) | 718.9 | 0.28 | 30.13 |
| SVD Blattmann et al. (2023a) | 857.2 | 1.1 | 23.95 |
| FreeNoise Qiu et al. (2024) | 1298.4 | 0 | 31.55 |
| OpenSora Zheng et al. (2024) | 1165.7 | 0.16 | 31.54 |
| OpenSoraPlan PKU-Yuan-Lab & Tuzhan-AI (2024) | 72.9 | 0.24 | 29.34 |
| StreamingT2V (*Ours*) | 52.3 | 0.04 | 31.73 |

best score by OpenSoraPlan). Likewise, our method achieves the second lowest SCuts score among all competitors. Only the methods FreeNoise achieves a slightly lower, perfect score. However, FreeNoise produces near-static videos (see also Fig. 5), leading automatically to low SCuts scores. OpenSoraPlan frequently produces scene cuts, leading to a 6 times higher SCuts score than our method. While SparseControl also follows a ControlNet approach, it leads to 100 times more scene cuts compared to StreamingT2V. This shows the advantage of our attentional CAM block over SparseControl, where the conditional frames need to be pad with zeros, so that inconsistency in the input lead to severe scene cuts.

Interestingly, all competing methods that incorporate CLIP image encodings are prone to misalignment (measured in low CLIP scores), *i.e*. SVD and DynamiCrafterXL and I2VGen-XL. We hypothesize that this is due to a domain shift; the CLIP image encoder is trained on natural images, but in an autoregressive setup, it is applied on generated images. With the help of our long-term memory, APM reminds the network about the domain of real images, as we use a fixed anchor frame, so that it does not degrade, and remains well-aligned to the textual prompt. Accordingly, StreamingT2V achieves the highest CLIP score among all evaluated methods.

**Qualitative Evaluation.** Finally, we present corresponding visual results on the test set in Fig. 5 (and in Sec. C). The high similarity of the frames depicted for competitors shows that all competing methods suffer from video stagnation, where the background and the camera is frozen, and nearly no object motion is generated. Our method is generating smooth and consistent videos without leading to standstill. I2VG, SVD, SparseCtrl, SEINE, OpenSoraPlan and DynamiCrafter-XL are prone to severe quality degradation, *e.g*. wrong colors and distorted frames, and inconsistencies, showing that their conditioning via CLIP image encoder and concatenation is too weak and heavily amplifies errors. In contrast, thanks to the more powerful CAM mechanism, StreamingT2V leads to smooth chunk transitions. APM conditions on a fixed anchor frame, so that StreamingT2V does not suffer from error accumulation.

## 6 CONCLUSION

In this paper, we tackled the challenge of generating long videos from textual prompts. We observed that all existing methods produce long videos either with temporal inconsistencies or severe stagnation up to standstill. To overcome these limitations, we carefully analysed an autoregressive pipeline build on top of a vanilla video diffusion model and proposed StreamingT2V, which incorporates short- and long-term dependency blocks to ensure smooth continuation of video chunks with high motion amount while maintaining scene and object features. We proposed a randomized blending approach that enables to use a video enhancer within the autoregressive process without temporal inconsistencies. Experimental results demonstrate that StreamingT2V outperforms competitors in terms of motion amount and temporal consistency, enabling the generation of long videos from text prompts without content stagnation.

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

## A    Appendix

## Appendix

This appendix complements our main paper with experiments, in which we further investigate the text-to-video generation quality of StreamingT2V, demonstrate even longer sequences than those assessed in the main paper, and provide additional information on the implementation of StreamingT2V and the experiments carried out.

In Sec. B, a user study is conducted on the test set, in which all text-to-video methods under consideration are evaluated by humans to determine the user preferences.

Sec. C supplements our main paper by additional qualitative results of StreamingT2V for very long video generation, and qualitative comparisons with competing methods.

In Sec. D, we present ablation studies to show the effectiveness of our proposed components CAM, APM and randomized blending.

In Sec. E, further training details, including hyperparameters used in StreamingT2V, and implementation details of our ablated models are provided.

Finally, Sec. F provides the prompts that compose our testset.

## B    User Study

We conduct a user study comparing our StreamingT2V method with prior work using the video results generated for the benchmark of Sec. 5.3. To remove potential biases, we resize and crop all videos to align them. The user study is structured as a one vs one comparison between our StreamingT2V method and competitors where participants are asked to answer three questions for each pair of videos:

- Which model has better motion?
- Which model has better text alignment?
- Which model has better overall quality?

We accept exactly one of the following three answers for each question: preference for the left model, preference for the right model, or results are considered equal. To ensure fairness, we randomize the order of the videos presented in each comparison, and the sequence of comparisons. Fig. 6 shows the preference score obtained from the user study as the percentage of votes devoted to the respective answer.

Across all comparisons to competing methods, StreamingT2V is significantly more often preferred than the competing method, which demonstrates that StreamingT2V clearly improves upon state-of-the-art for long video generation. For instance in motion quality, as the results of StreamingT2V are non-stagnating videos, temporal consistent and possess seamless transitions between chunks, $65\%$ of the votes were preferring StreamingT2V, compared to $17\%$ of the votes preferring SEINE.

Competing methods are much more affected by quality degradation over time, which is reflected in the preference for StreamingT2V in terms of *text alignment* and *overall quality*.

## C    Qualitative Results

Complementing our visual results shown in the main paper (see Fig. 5), we present additional qualitative results of StreamingsT2V on our test set on very long video generation, and further qualitative comparisons to prior works on 240 frames.

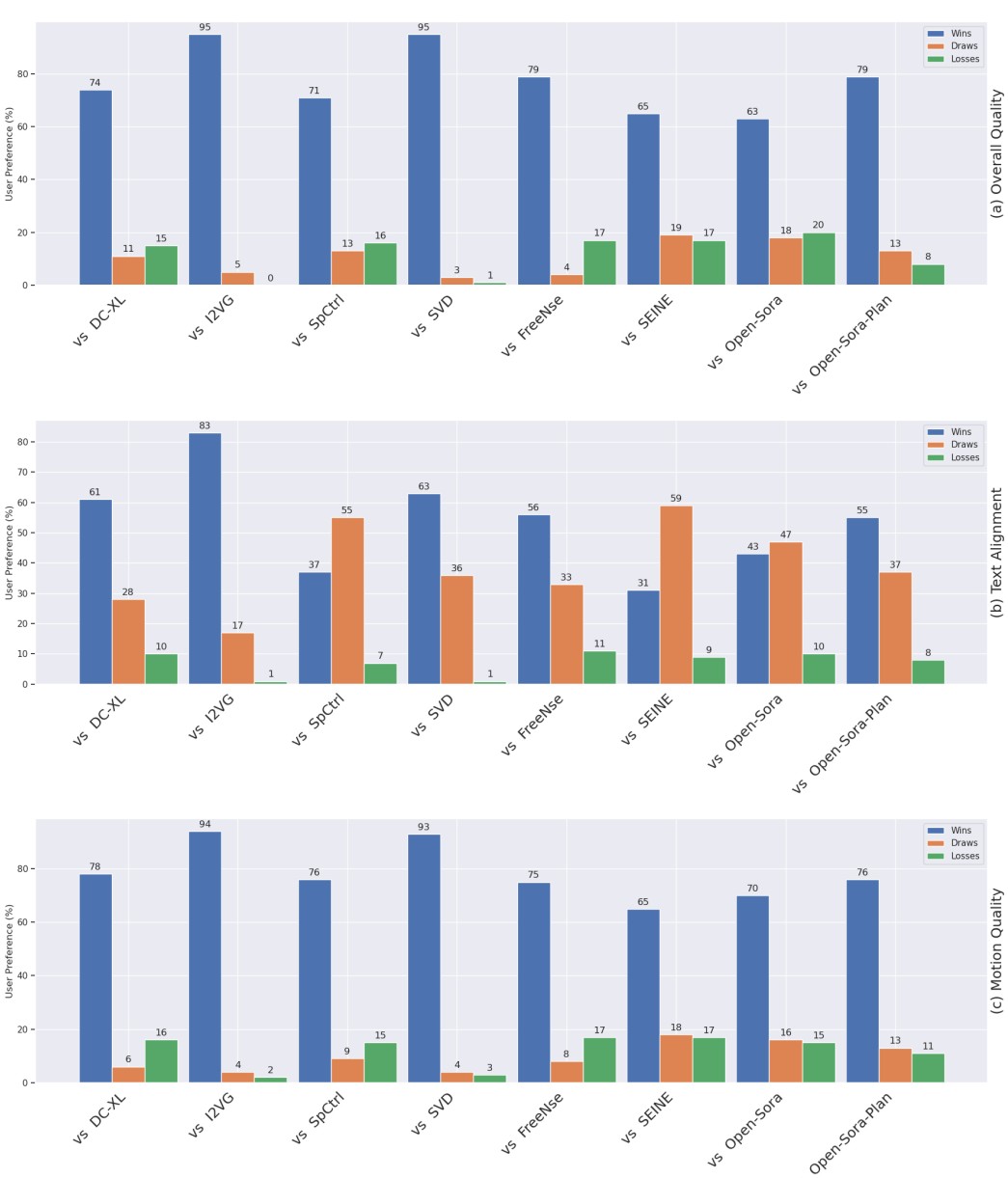

Figure 6: We conduct a user study, asking humans to assess the test set results of Sec. 5.3 in a one-to-one evaluation, where for any prompt of the test set and any competing method, the results of the competing method have to be compared with the corresponding results of our StreamingT2V method. For each comparison of our method to a competing method, we report the relative of number votes that prefer StreamingT2V (*i.e.* wins), that prefer the competing method (*i.e.* losses), and that consider results from both methods as equal (*i.e.* draws).

## C.1 VERY LONG VIDEO GENERATION

Supplementing our main paper, we show that StreamingT2V can be used for very long video generation. To this end, we generate and show videos consisting of 1200 frames, thus spanning 2 minutes, which is 5 times longer than the ones produced for the experiments in our main paper. Fig. 7 show these text-to-video results of StreamingT2V for different actions, *e.g. dancing*, *running*, or *camera moving*, and different characters like *bees* or *jellyfish*. We can observe that scene and object features are kept across each video generation (see *e.g.* Fig. 7(a)&(e)), thanks to our proposed APM module.

Our proposed CAM module ensures that generated videos are temporally smooth, with seamless transitions between video chunks, and not stagnating (see *e.g.* Fig. 7(f)&(k)).

(a) People dancing in room filled with fog and colorful lights

(b) Camera moving in a wide bright ice cave

(c) Marvel at the diversity of bee species

(d) Dive into the depths of the ocean: explore vibrant coral reefs

(e) Venture into the kelp forests: weave through towering underwater forests

(f) Experience the dance of jellyfish: float through mesmerizing swarms of jellyfish

(g) Enter the realm of ice caves: venture into frozen landscapes

(h) Wide shot of battlefield, stormtroopers running at night, smoke, fires and smokes

(i) Witness the wonders of sea caves

(j) Camera moving around vast deserts, where dunes stretch endlessly into the horizon

(k) Enter the fascinating world of bees: explore the intricate workings of a beehive

Figure 7: Qualitative results of StreamingT2V for different prompts. Each video has 1200 frames.

## C.2 MORE QUALITATIVE EVALUATIONS.

The visual comparisons shown in Fig. 8, 9, 10, 11 demonstrate that StreamingT2V significantly excels the generation quality of all competing methods. StreamingT2V shows non-stagnating videos with good motion quality, in particular seamless transitions between chunks and temporal consistency.

Videos generated by DynamiCrafter-XL eventually possess severe image quality degradation. For instance, we observe in Fig. 8 eventually wrong colors at the beagle's face and the background pattern heavily deteriorates. The quality degradation also heavily deteriorates the textual alignment (see the result of DynamiCrafter-XL in Fig. 10). Across all visual results, the method SVD is even more susceptible to these issues.

The methods SparseControl and FreeNoise eventually lead to almost stand-still, and are thus not able to perform the action described in a prompt, *e.g.* "zooming out" in Fig. 11. Likewise, also SEINE is not following this camera instructions (see Fig. 11).

OpenSora is mostly not generating any motion, leading either to complete static results (Fig. 8), or some image warping without motion (Fig. 10). OpenSoraPlan is loosing initial object details and suffers heavily from quality degradation through the autoregressive process, *e.g.* the dog is hardly recognizable at the of the video generation (see Fig. 8), showing again that a sophisticated conditioning mechanism is necessary.

I2VGen-XL shows low motion amount, and eventually quality degradation, leading eventually to frames that are weakly aligned to the textual instructions.

We further analyse visually the chunk transitions using an X-T slice visualization in Fig. 12. We can observe that StreamingT2V leads to smooth transitions. In contrast, we observe that conditioning via CLIP or concatenation may lead to strong inconsistencies between chunks.

# D ABLATION STUDIES

To assess the importance of our proposed components, we conduct several ablation studies on a randomly sampled set of 75 prompts from our validation set that we used during training.

Specifically, we compare CAM against established conditioning approaches in Sec. D.1, analyse the impact of our long-term memory APM in Sec. D.2, and ablate on our modifications for the video enhancer in Sec. D.3.

## D.1 CONDITIONAL ATTENTION MODULE.

To analyse the importance of CAM, we compare CAM (w/o APM) with two baselines (baseline details in Sec. D.1.1): (i) Connect the features of CAM with the skip-connection of the UNet via zero convolution, followed by addition. We zero-pad the condition frame and concatenate it with a frame-indicating mask to form the input for the modified CAM, which we denote as Add-Cond. (ii) We append the conditional frames and a frame-indicating mask to input of Video-LDM's Unet along the channel dimension, but do not use CAM, which we denote as Conc-Cond. We train our method with CAM and the baselines on the same dataset. Architectural details (including training) of these baselines are provided in the appendix.

We obtain an SCuts score of 0.24, 0.284 and 0.03 for Conc-Cond, Add-Cond and Ours (w/o APM), respectively. This shows that the inconsistencies in the input caused by the masking leads to frequent inconsistencies in the generated videos and that concatenation to the Unet's input is a too weak conditioning. In contrast, our CAM generates consistent videos with a SCuts score that is 88% lower than the baselines.

### D.1.1 ABLATION MODELS

For the ablation of CAM, we considered two baselines that we compare with CAM. Here we provide additional implementation details of these baselines.

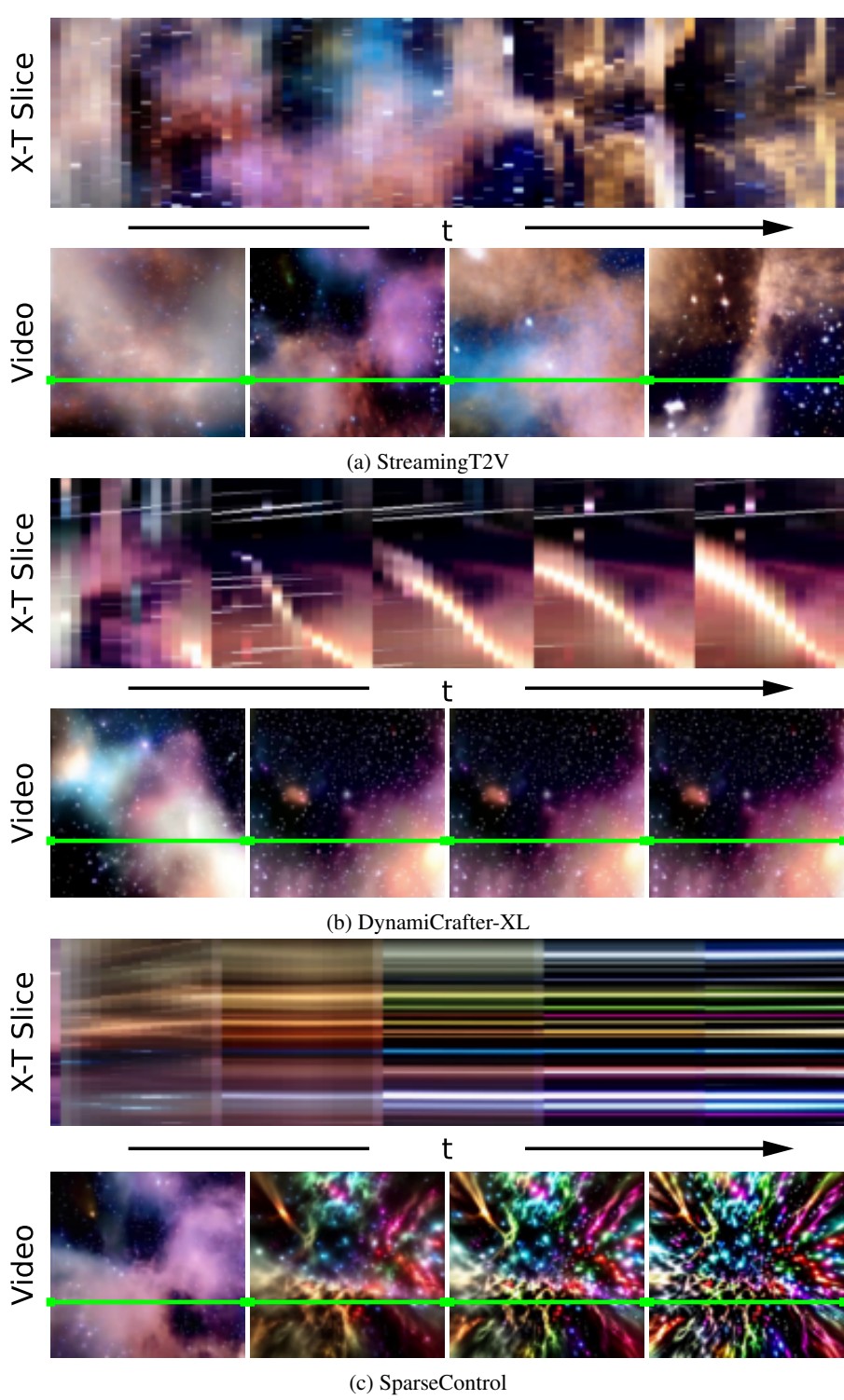

Figure 12: Visual comparison of SparseControl, DynamiCrafter-XL and StreamingT2V. All text-to-video results are generated using the same prompt. The X-T slice visualization shows that DynamiCrafter-XL and SparseControl suffer from severe chunk inconsistencies and repetitive motions. In contrast, our method shows seamless transitions and evolving content.

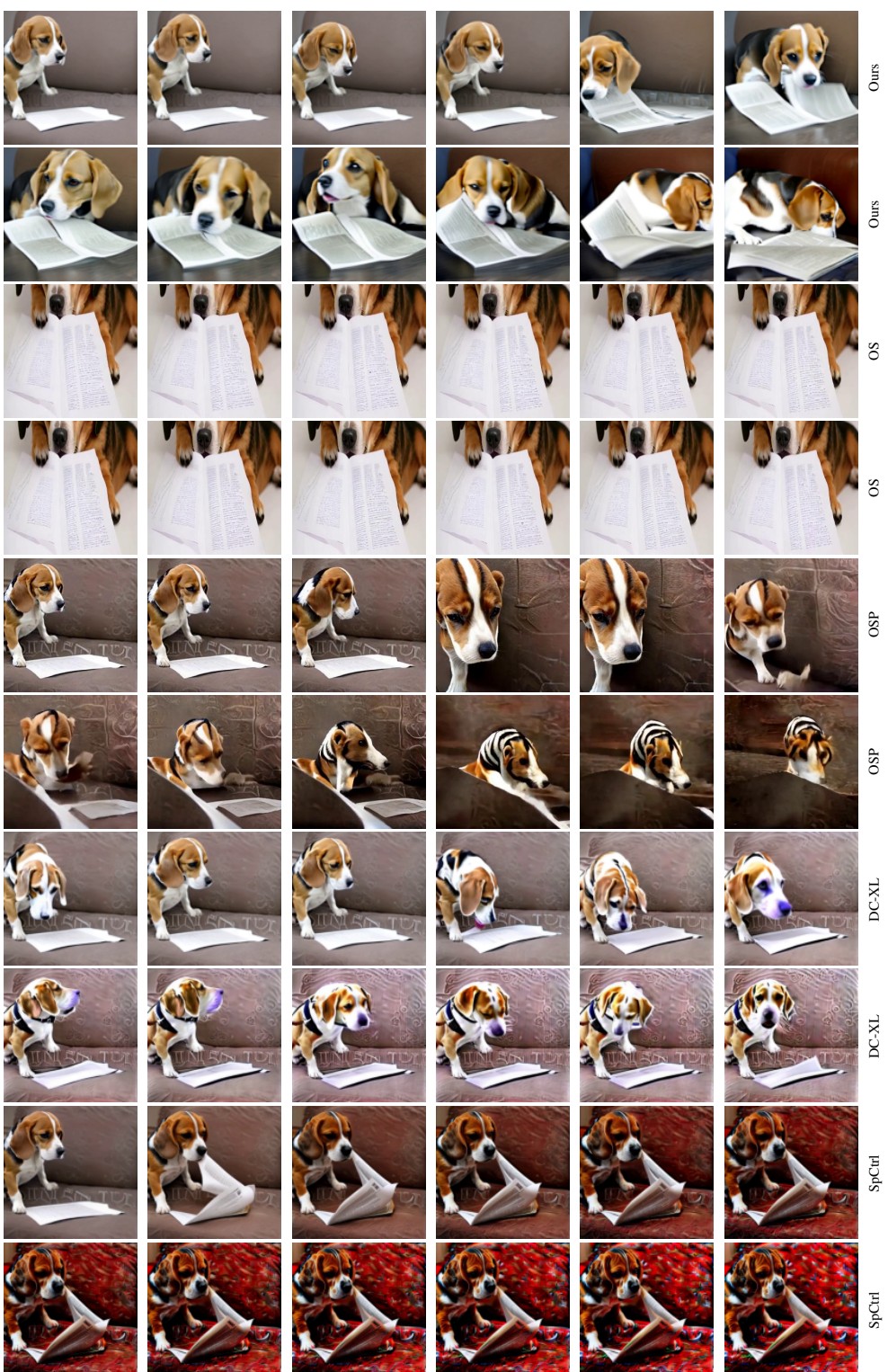

Figure 8: Video generation for the prompt "*A beagle reading a paper*", using StreamingT2V and competing methods. For each method, the image sequence of its first row is continued by the image in the leftmost column of the following row.

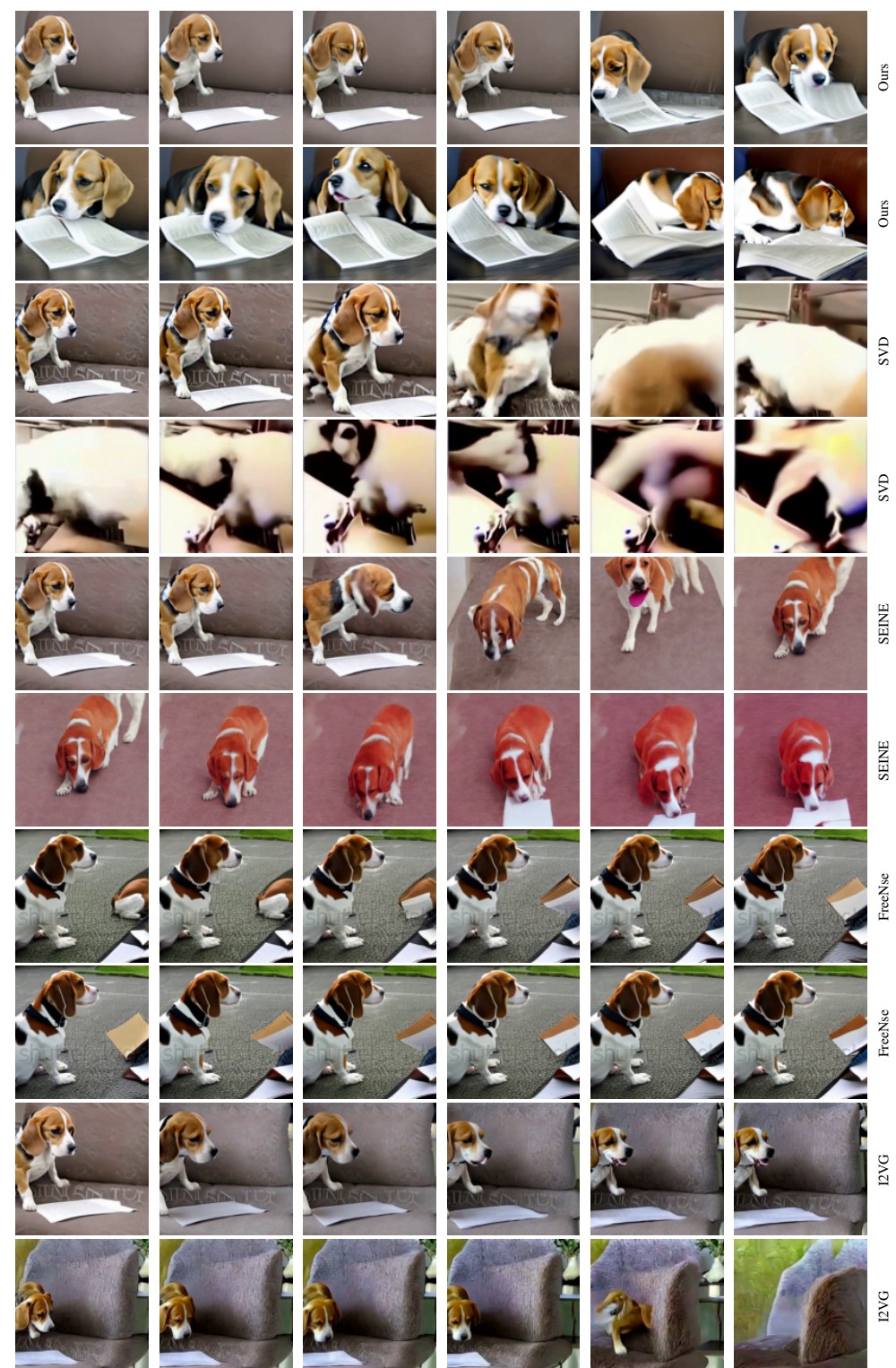

Figure 9: Video generation for the prompt "*A beagle reading a paper*", using StreamingT2V and competing methods. For each method, the image sequence of its first row is continued by the image in the leftmost column of the following row.

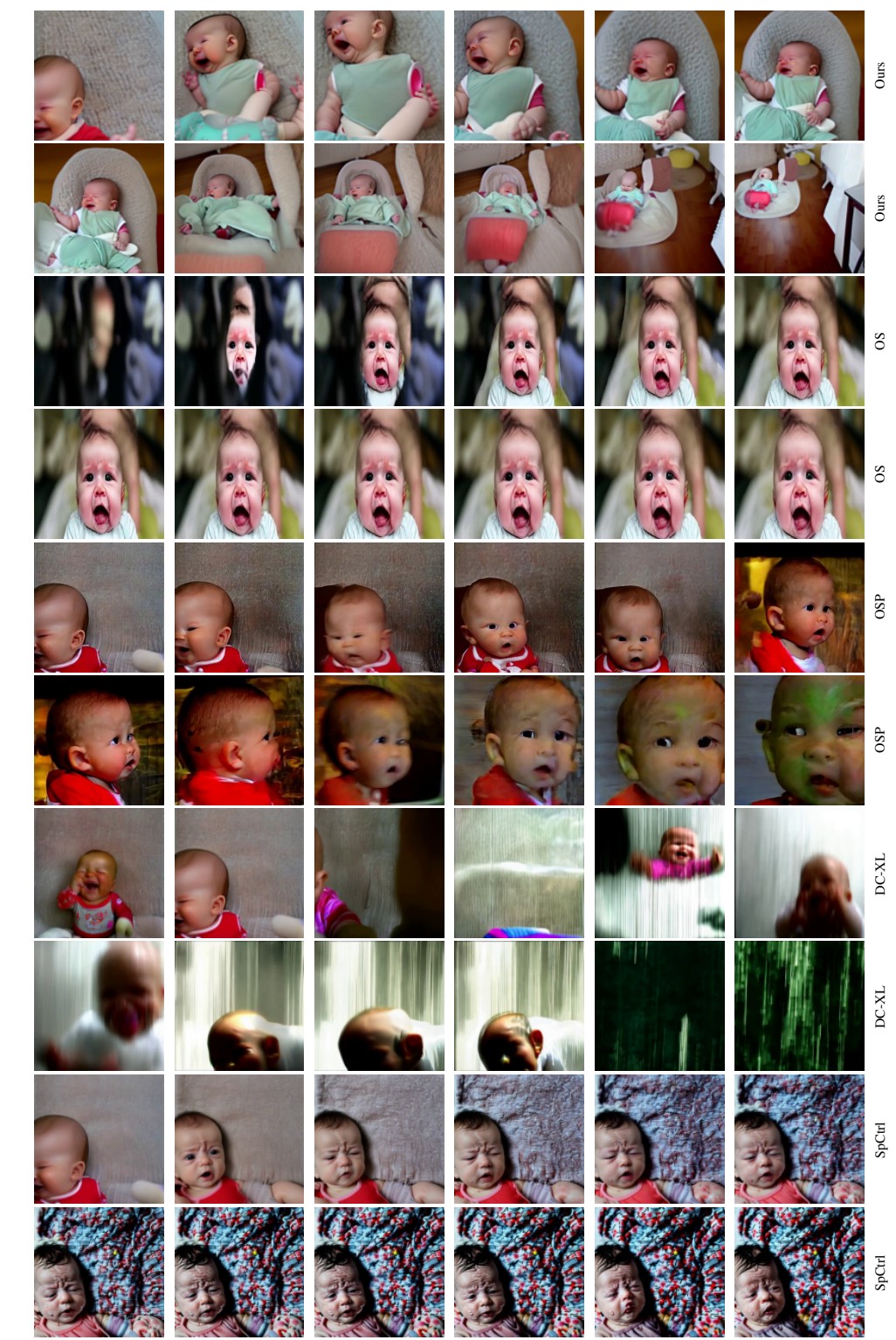

Figure 10: Video generation for the prompt "*Camera is zooming out and the baby starts to cry*", using StreamingT2V and competing methods. For each method, the image sequence of its first row is continued by the image in the leftmost column of the following row.

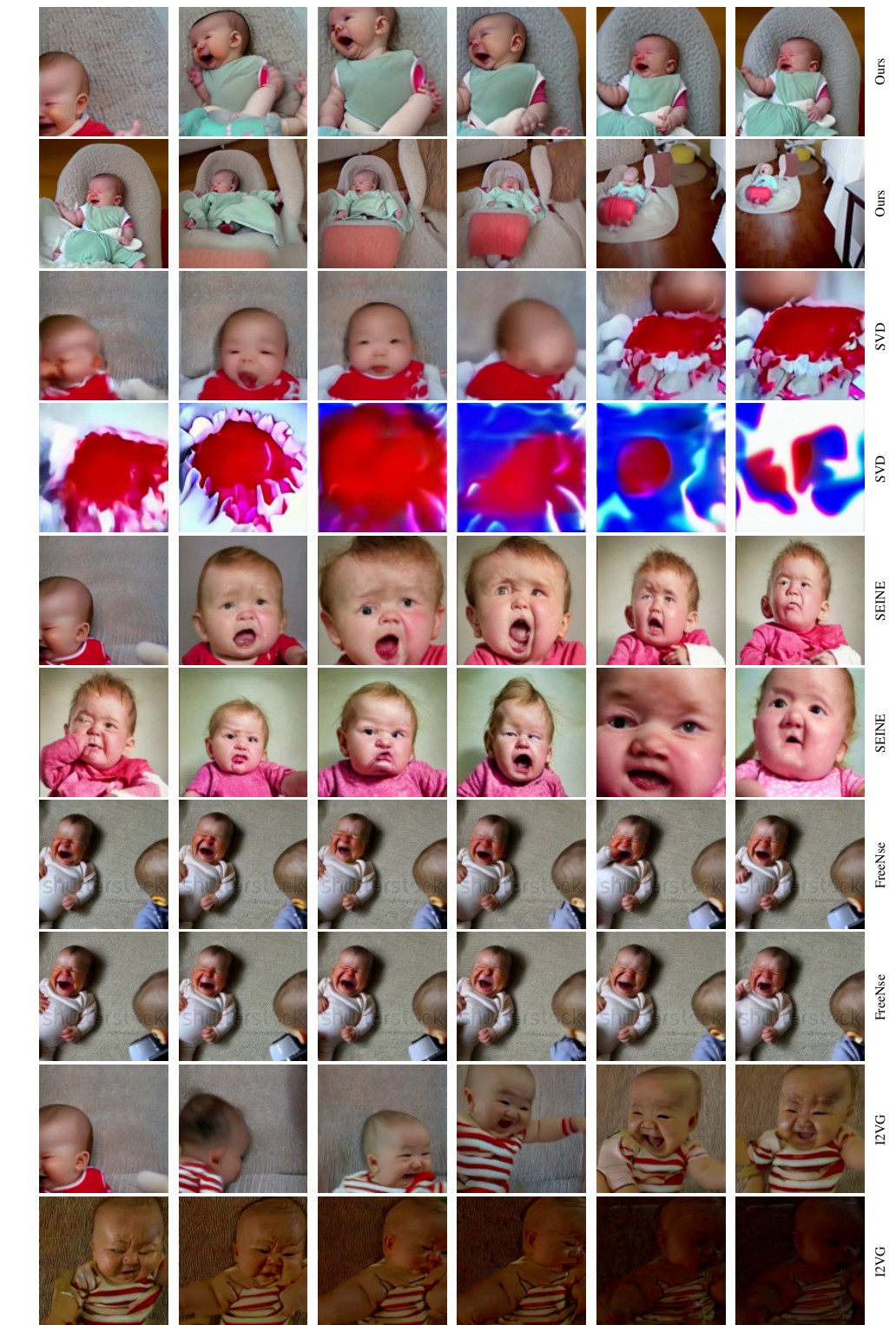

Figure 11: Video generation for the prompt "*Camera is zooming out and the baby starts to cry*", using StreamingT2V and competing methods. For each method, the image sequence of its first row is continued by the image in the leftmost column of the following row.

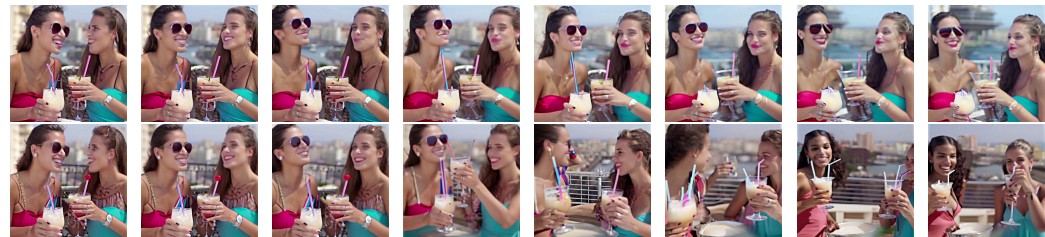

(a) Young caucasian female couple drinking cocktails and smiling on terrace in havana, cuba. girls, teens, teenagers, women

Figure 13: Top row: CAM+APM, Bottom row: CAM. The figure shows that using long-term information via APM helps to keep identities (e.g. the face of the left woman) and scene features, e.g. the dresses or arm clock.

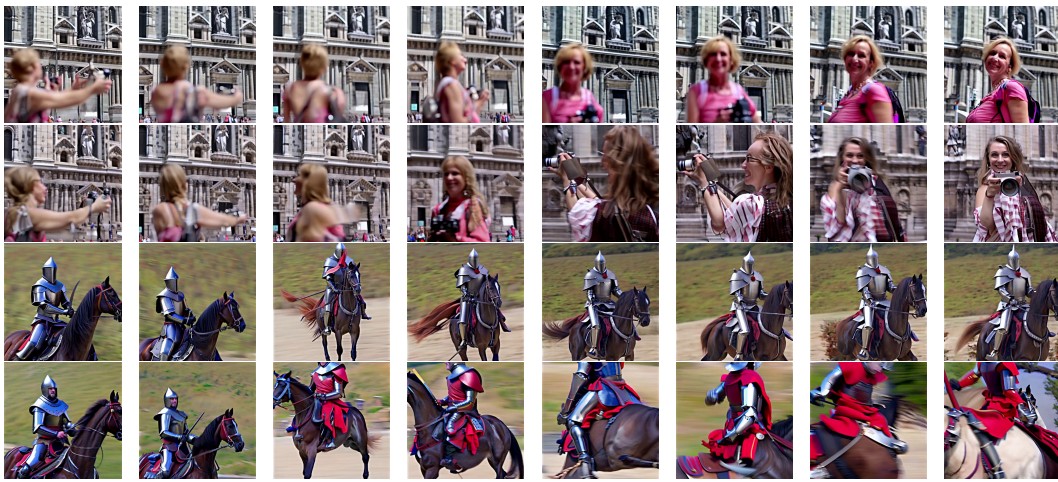

Figure 14: Ablation study on the APM module. Top row is generated from StreamingT2V, bottom row is generated from StreamingT2V w/o APM.

The ablated model *Add-Cond* applies to the features of CAM (*i.e.* the outputs of the encoder and middle layer of the ControlNet part in Fig. 3) zero-convolution, and uses addition to fuse it with the features of the skip-connection of the UNet (similar to ControlNet Zhang et al. (2023b)) (see Fig. 16). We provide here additional details to construct this model. Given a video sample $\mathcal{V} \in \mathbb{R}^{F \times H \times W \times 3}$ with $F = 16$ frames, we construct a mask $M \in \{0, 1\}^{F \times H \times W \times 3}$ that indicates which frame we use for conditioning, *i.e.* $M^f[i, j, k] = M^f[i', j', k']$ for all frames $f = 1, \ldots, F$ and for all $i, j, k, i', j', k'$. We require that exactly $F - F_{\text{cond}}$ frames are masked, *i.e.*

$$\sum_{f=1}^{F} M^f[i, j, k] = F - F_{\text{cond}}, \text{ for all } i, j, k. \tag{10}$$

We concatenate $[\mathcal{V} \odot M, M]$ along the channel dimension and use it as input for the image encoder $\mathcal{E}_{\text{cond}}$, where $\odot$ denotes element-wise multiplication.

During training, we randomly set the mask $M$. During inference, we set the mask for the first 8 frames to zero, and for the last 8 frames to one, so that the model conditions on the last 8 frames of the previous chunk.

For the ablated model *Conc-Cond*, we start from our Video-LDM's UNet, and modify its first convolution. Like for *Add-Cond*, we consider a video $\mathcal{V}$ of length $F = 16$ and a mask $M$ that encodes which frames are overwritten by zeros. Now the Unet takes $[z_t, \mathcal{E}(\mathcal{V}) \odot M, M]$ as input, where we concatenate along the channel dimension. As with *Add-Cond*, we randomly set $M$ during training

so that the information of 8 frames is used, while during inference, we set it such that the last 8 frames of the previous chunk are used. Here $\mathcal{E}$ denotes the VQ-GAN encoder (see Sec. 3).

## D.2 APPEARANCE PRESERVATION MODULE

We analyse the impact of utilizing long-term memory in the context of long video generation.

Fig. 13 and Fig. 14 show that long-term memory greatly helps keeping the object and scene features across autoregressive generations. Thanks to the usage of long-term information via our proposed APM module, identity and scene features are preserved throughout the video. For instance, the face of the woman in Fig. 14 (including all its tiny details) are consistent[1] across the video generation. Also, the style of the jacket and the bag are correctly generated throughout the video. Without having access to a long-term memory, these object and scene features are changing over time.

This is also supported quantitatively. We utilize a person re-identification score to measure feature preservation (definition in Sec. D.2.1), and obtain scores of 93.42 and 94.95 for Ours w/o APM, and Ours, respectively. Our APM module thus improves the identity/appearance preservation. Also the scene information is better kept, as we observe an image distance score in terms of LPIPS Zhang et al. (2018) of 0.192 and 0.151 for Ours w/o APM and Ours, respectively. We thus have an improvement in terms of scene preservation of more than 20% when APM is used.

### D.2.1 MEASURING FEATURE PRESERVATION.

We employ a person re-identification score as a proxy to measure feature preservation. To this end, let $P_n = \{p_i^n\}$ be all face patches extracted from frame $n$ using an off-the-shelf head detector Schroff et al. (2015) and let $F_i^n$ be the corresponding facial feature of $p_i^n$, which we obtain from an off-the-shelf face recognition network Schroff et al. (2015). Then, for frame $n$, $n_1 := |P_n|$, $n_2 := |P_{n+1}|$, we define the re-id score re-id$(n)$ for frame $n$ as

$$\text{re-id}(n) := \begin{cases} \max_{i,j} \cos \Theta(F_i^n, F_j^{n+1}), & n_1 > 0 \ \& \ n_2 > 0. \\ 0 & \text{otherwise.} \end{cases} \tag{11}$$

where $\cos \Theta$ is the cosine similarity. Finally, we obtain the re-ID score of a video by averaging over all frames, where the two consecutive frames have face detections, *i.e.* with $m := |\{n \in \{1, .., N\} : |P_n| > 0\}|$, we compute the weighted sum:

$$\text{re-id} := \frac{1}{m} \sum_{n=1}^{N-1} \text{re-id}(n), \tag{12}$$

where $N$ denotes the number of frames.

## D.3 RANDOMIZED BLENDING.

We assess our randomized blending approach by comparing against two baselines. (B) enhances each video chunk independently, and (B+S) uses shared noise for consecutive chunks, with an overlap of 8 frames, but not randomized blending. We compute per sequence the standard deviation of the optical flow magnitudes between consecutive frames and average over all frames and sequences, which indicates temporal smoothness. We obtain the scores 8.72, 6.01 and 3.32 for B, B+S, and StreamingT2V, respectively. Thus, noise sharing improves chunk consistency (by 31% vs B), but it is significantly further improved by randomized blending (by 62% vs B).

These findings are supported visually. Fig. 15 shows ablated results on our randomized blending approach. From the X-T slice visualizations we can see that the randomized blending leads to smooth chunk transitions, confirming our observations and quantitative evaluations. In contrast, when naively concatenating enhanced video chunks, or using shared noise, the resulting videos possess visible inconsistencies between chunks.

---

[1]The background appears to have changed. However, please note that the camera is rotating so that a different area behind the two woman becomes visible, so that the background change is correct.

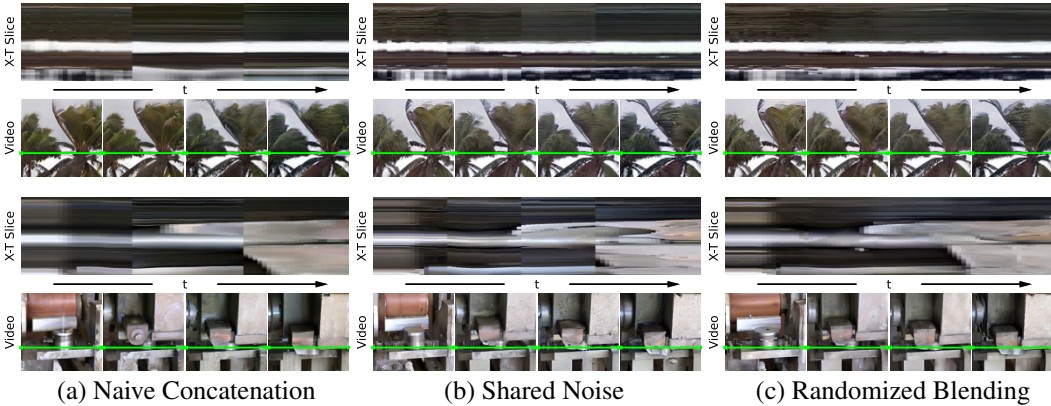

| (a) Naive Concatenation | (b) Shared Noise | (c) Randomized Blending |

Figure 15: Ablation study on our video enhancer improvements. The X-T slice visualization shows that randomized blending leads to smooth chunk transitions, while both baselines have clearly visible, severe inconsistencies between chunks.

## E  IMPLEMENTATION DETAILS

We provide additional details regarding the training of StreamingT2V and further implementation details.

### E.1  TRAINING DETAILS

In order to train the APM module, we randomly sample an anchor frame out of the first 16 frames. For the conditioning and denoising, we use the frames $17 - 24$ and $17 - 32$, respectively. This aligns training with inference, where there is a large time gap between the conditional frames and the anchor frame. In addition, by randomly sampling an anchor frame, the model can leverage the CLIP information only for the extraction of high-level semantic information, as we do not provide a frame index to the model.

### E.2  STREAMING T2V STAGE

For the StreamingT2V stage, we use classifier free guidance Ho & Salimans (2021); Esser et al. (2023) from text and the anchor frame. More precisely, let $\epsilon_\theta(x_t, t, \tau, a)$ denote the noise prediction in the StreamingT2V stage for latent code $x_t$ at diffusion step $t$, text $\tau$ and anchor frame $a$. For text guidance and guidance by the anchor frame, we introduce weights $\omega_{\text{text}}$ and $\omega_{\text{anchor}}$, respectively. Let $\tau_{\text{null}}$ and $a_{\text{null}}$ denote the empty string, and the image with all pixel values set to zero, respectively. Then, we obtain the multi-conditioned classifier-free-guided noise prediction $\hat{\epsilon}_\theta$ (similar to DynamiCrafter-XL Xing et al. (2023)) from the noise predictor $\epsilon$ via

$$\hat{\epsilon}_\theta(x_t, t, \tau, a) = \epsilon_\theta(x_t, t, \tau_{\text{null}}, a_{\text{null}}) + \omega_{\text{text}}\big(\epsilon_\theta(x_t, t, \tau, a_{\text{null}}) - \epsilon_\theta(x_t, t, \tau_{\text{null}}, a_{\text{null}})\big)$$
$$+ \omega_{\text{anchor}}\big(\epsilon_\theta(x_t, t, \tau, a) - \epsilon_\theta(x_t, t, \tau, a_{\text{null}})\big). \tag{13}$$

We then use $\hat{\epsilon}_\theta$ for denoising. In our experiments, we set $\omega_{\text{text}} = \omega_{\text{anchor}} = 7.5$. During training, we randomly replace $\tau$ with $\tau_{\text{null}}$ with $5\%$ likelihood, the anchor frame $a$ with $a_{\text{null}}$ with $5\%$ likelihood, and we replace at the same time $\tau$ with $\tau_{\text{null}}$ and $a$ with $a_{\text{null}}$ with $5\%$ likelihood.

Additional hyperparameters for the architecture, training and inference of the Streaming T2V stage are presented in Tab. 12, where *Per-Pixel Temporal Attention* refers to the attention module used in CAM (see Fig. 3).

## F  TEST SET PROMPTS

1. A camel resting on the snow field.
2. Camera following a pack of crows flying in the sky.

Table 12: Hyperparameters of Streaming T2V Stage. Additional architectural hyperparameters are provided by the Modelsope report Wang et al. (2023b).

| **Per-Pixel Temporal Attention** | |
|---|---|
| Sequence length Q | 16 |
| Sequence length K,V | 8 |
| Token dimensions | 320, 640, 1280 |
| **Appearance Preservation Module** | |
| CLIP Image Embedding Dim | 1024 |
| CLIP Image Embedding Tokens | 1 |
| MLP hidden layers | 1 |
| MLP inner dim | 1280 |
| MLP output tokens | 16 |
| MLP output dim | 1024 |
| 1D Conv input tokens | 93 |
| 1D Conv output tokens | 77 |
| 1D Conv output dim | 1024 |
| Cross attention sequence length | 77 |
| **Training** | |
| Parametrization | $\epsilon$ |
| **Diffusion Setup** | |
| Diffusion steps | 1000 |
| Noise scheduler | Linear |
| $\beta_0$ | 0.0085 |
| $\beta_T$ | 0.0120 |
| **Sampling Parameters** | |
| Sampler | DDIM |
| Steps | 50 |
| $\eta$ | 1.0 |
| $\omega_{\text{text}}$ | 7.5 |
| $\omega_{\text{anchor}}$ | 7.5 |

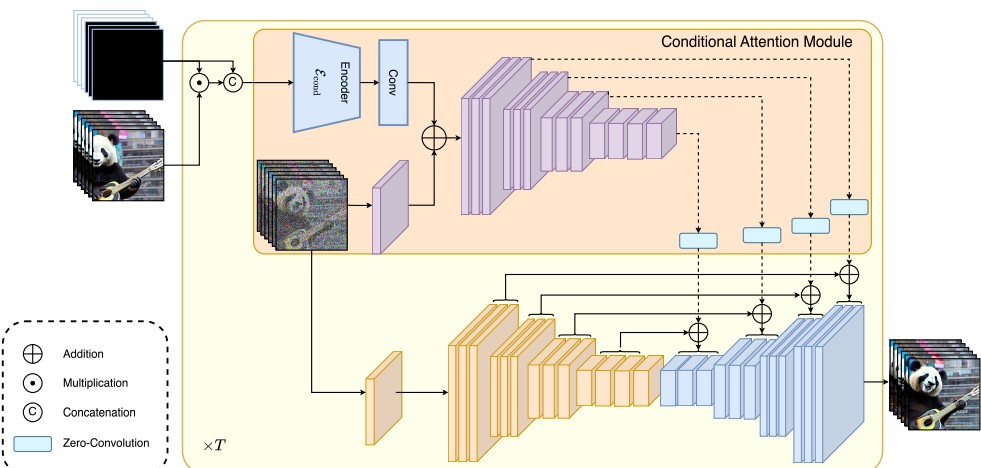

Figure 16: Illustration of the Add-Cond baseline, which is used in Sec. D.1.

3. A knight riding on a horse through the countryside.

4. A gorilla eats a banana in Central Park.

5. Men walking in the rain.

6. Ants, beetles and centipede nest.

7. A squirrel on a table full of big nuts.

8. Close flyover over a large wheat field in the early morning sunlight.

9. A squirrel watches with sweet eyes into the camera.

10. Santa Claus is dancing.

11. Chemical reaction.

12. Camera moving in a wide bright ice cave, cyan.

13. Prague, Czech Republic. Heavy rain on the street.

14. Time-lapse of stormclouds during thunderstorm.

15. People dancing in room filled with fog and colorful lights.

16. Big celebration with fireworks.

17. Aerial view of a large city.

18. Wide shot of battlefield, stormtroopers running at night, fires and smokes and explosions in background.

19. Explosion.

20. Drone flythrough of a tropical jungle with many birds.

21. A camel running on the snow field.

22. Fishes swimming in ocean camera moving.

23. A squirrel in Antarctica, on a pile of hazelnuts cinematic.

24. Fluids mixing and changing colors, closeup.

25. A horse eating grass on a lawn.

26. The fire in the car is extinguished by heavy rain.

27. Camera is zooming out and the baby starts to cry.

28. Flying through nebulas and stars.

29. A kitten resting on a ball of wool.

30. A musk ox grazing on beautiful wildflowers.

31. A hummingbird flutters among colorful flowers, its wings beating rapidly.

32. A knight riding a horse, pointing with his lance to the sky.

33. steampunk robot looking at the camera.

34. Drone fly to a mansion in a tropical forest.

35. Top-down footage of a dirt road in forest.

36. Camera moving closely over beautiful roses blooming time-lapse.

37. A tiger eating raw meat on the street.

38. A beagle looking in the Louvre at a painting.

39. A beagle reading a paper.

40. A panda playing guitar on Times Square.

41. A young girl making selfies with her phone in a crowded street.

42. Aerial: flying above a breathtaking limestone structure on a serene and exotic island.

43. Aerial: Hovering above a picturesque mountain range on a peaceful and idyllic island getaway.

44. A time-lapse sequence illustrating the stages of growth in a flourishing field of corn.

45. Documenting the growth cycle of vibrant lavender flowers in a mesmerizing time-lapse.

46. Around the lively streets of Corso Como, a fearless urban rabbit hopped playfully, seemingly unfazed by the fashionable surroundings.

47. Beside the Duomo's majestic spires, a fearless falcon soared, riding the currents of air above the iconic cathedral.

48. A graceful heron stood poised near the reflecting pools of the Duomo, adding a touch of tranquility to the vibrant surroundings.

49. A woman with a camera in hand joyfully skipped along the perimeter of the Duomo, capturing the essence of the moment.

50. Beside the ancient amphitheater of Taormina, a group of friends enjoyed a leisurely picnic, taking in the breathtaking views.

