# OpenReview forum: "StreamingT2V: Consistent, Dynamic, and Extendable Long Video Generation from Text"
_ICLR.cc/2025/Conference — ICLR 2025 Conference Withdrawn Submission_

### Official Review · Reviewer_3hpQ · 2024-10-30

**Soundness:** 3
**Presentation:** 2
**Contribution:** 3
**Rating:** 5
**Confidence:** 5

**Summary:**

This paper presents a long video generation framework from single text prompt. The main contribution is the proposed conditional attention module (CAM) and appearance preservation module (APM) for temporal consistent long video generation.

**Strengths:**

-An autoregressive long video generation framework is designed, which is novel, and shows stable video quality.

**Weaknesses:**

-I'm wondering the necessity of generating very long clip with only one short caption. In example videos provided in the supplementary material, it seems the content of video is limited to a very narrow domain with little variation, due to the design of APM. It is not suitable for very long video generation.

-In line 299, "...fuse x with output of the first temporal transformer block of CAM." Just curious about the fusion here, as in Figure 3, x seems to be added with the noised input after one encoding layer. Can this encoding layer described as the first temporal transformer block of CAM? As generally, the first block of CAM should have the skip connections to decoding part.

-In line 417, the mean warp error W(V) is the average squared L2 pixel distance from a frame to its warped subsequent frame. So is it computed by calculating the warp error between anchor frame and all other frames? Or between two consecutive frames? What's the definition of warp error?

-The quantitative comparison only includes the long video generation quality evaluation, lacking the common metric evaluation, such as FVD, LPIPS. Also lacks the evaluation on common datasets like MSRVTT and UCF101.

**Questions:**

Please see the weaknesses.

---

### Official Review · Reviewer_hVew · 2024-10-31

**Soundness:** 2
**Presentation:** 2
**Contribution:** 2
**Rating:** 5
**Confidence:** 3

**Summary:**

The paper presents a novel text-to-video diffusion model aimed at generating long videos. Addressing the challenge of abrupt transitions in extended videos, the model incorporates three key mechanisms: a Conditional Attention Module (CAM) for smooth short-term transitions, an Appearance Preservation Module (APM) to maintain scene consistency, and a randomized blending technique for refining generated videos.

**Strengths:**

1.The proposed autoregressive approach effectively leverages both short-term and long-term dependencies, facilitating the seamless creation of extended video content. This method adeptly addresses the challenges associated with producing longer video sequences by ensuring smooth transitions and continuity.

2.Through the integration of the Conditional Attention Module (CAM) and the Appearance Preservation Module (APM), the model ensures that generated videos exhibit natural continuity and maintain consistent scene and object characteristics across their entire length.

**Weaknesses:**

1. CAM design :  In the W.A.L.T[1] method, a very straightforward auto-regressive generation approach is provided for frame prediction tasks, where past generated frames are used as conditions to guide the generation of subsequent video content through the standard classifier-free guidance method. Can the authors explain why this approach was not adopted in the design of the CAM module, but rather a ControlNet method was used? Additionally, can the authors provide a comparison of the FVD metrics for the CAM and WALT frame prediction methods on the UCF-101 or K600 datasets?

2. Training details are missing: Can the authors provide details related to the training data?

3. Evaluation is a bit weak: Can the authors provide a comparison of FVD with other methods on the UCF-101 or K600 datasets?

---------
[1].Gupta, Agrim, Lijun Yu, Kihyuk Sohn, Xiuye Gu, Meera Hahn, Li Fei-Fei, Irfan Essa, Lu Jiang, and José Lezama. "Photorealistic video generation with diffusion models." arXiv preprint arXiv:2312.06662 (2023).

**Questions:**

1. Anchor frame influence: During the training and sampling stages, anchor frames are randomly sampled. How significant is the impact of choosing different anchor frames on the final video generation? Why can't all frames from the first chunk be used as anchor frames to guide generation?

---

### Official Review · Reviewer_dyaf · 2024-11-03

**Soundness:** 3
**Presentation:** 2
**Contribution:** 2
**Rating:** 5
**Confidence:** 4

**Summary:**

The paper proposes a streamable text-to-video method which can generate up to 2 minutes or longer videos with seamless transitions. Three innovative methods are proposed to ensure the long video consistency and overall quality. Firstly, conditional attention module injects previous-chunk information into the pre-trained video diffusion model to ensure smooth transitions between chunks. Secondly, the CLIP feature of the first frame is injected to the video diffusion model to ensure a coherent scene and object appearance within the whole video. Thirdly, a randomized blending approach is introduced to address inconsistent transitions caused by noise mismatch within the video enhancer's denoising process. A novel motion aware warp error metric is proposed to assess both motion amount and consistency. Experiments are conducted to evaluate the proposed method qualitatively and quantitatively.

**Strengths:**

1. The generated videos are sufficiently long, natural and with relatively large motion. The quantitative performance outperforms existing methods.
2. The paper identifies a noise mismatch problem when enhancing long videos using chunk-wise SDEdit, and proposes a randomized blending method to address this problem.

**Weaknesses:**

1. The novelty is limited. Firstly, generating subsequent frames with the condition of previous frame chunks has already been explored [1]. Secondly, the appearance preservation module (APM) in this paper is much like the anchored conditioning method in ART-V [2].
2. The paper states that the training data is collected from publicly available sources, but the corresponding URLs or papers are provided or mentioned. Please provide the URLs or citations for these sources.
3. Comparisons on general video quality benchmarks are missing, such as FVD and FID on MSR-VTT or UCF datasets.
4. The paper is not well written. The formatting issues make the paper unfrendly to read, e.g. it is better to use brackets when citing papers; Table 6 exceeds the width limit.

[1] Gao, Kaifeng, et al. "ViD-GPT: Introducing GPT-style Autoregressive Generation in Video Diffusion Models." arXiv preprint arXiv:2406.10981 (2024).

[2] Weng, Wenming, et al. "ART-V: Auto-Regressive Text-to-Video Generation with Diffusion Models." Proceedings of the IEEE/CVF Conference on Computer Vision and Pattern Recognition. 2024.

**Questions:**

About video length stress test. In auto-regressive video generation, there exists error accumulation problem, i.e. the generated frames have different distribution from the training data distribution, which makes the subsequently generated frames degrades further. How does StreamingT2V address the error accumulation problem? What is the upper-bound generation length of this model?

---

### Official Review · Reviewer_RPjY · 2024-11-04

**Soundness:** 3
**Presentation:** 2
**Contribution:** 2
**Rating:** 6
**Confidence:** 5

**Summary:**

This paper presents StreamingT2V, a method for generating high-quality, extended videos from text prompts, specifically addressing the challenge of ensuring smooth transitions in long-form content. Existing methods often struggle with abrupt cuts in longer videos. In contrast, StreamingT2V introduces three core components: (i) the Conditional Attention Module (CAM), a short-term memory mechanism that aligns each generated segment with its predecessor for seamless transitions; (ii) the Appearance Preservation Module (APM), a long-term memory unit that retains key features from the initial frames to maintain scene consistency; and (iii) a randomized blending technique that enables a video enhancer to be applied autoregressively, ensuring coherence over extended durations. Experiments demonstrate that StreamingT2V achieves high levels of motion and continuity, outperforming other models that tend to stagnate during prolonged autoregressive use.

**Strengths:**

1. The abstract and introduction repeatedly emphasize that the Appearance Preservation Module (APM)  ensures the natural continuity of object characteristics in generated videos. However, the paper does not provide metrics similar to CLIP-I to quantify the preservation of subject consistency.
2. When considering long video generation, users typically seek dynamic visuals rather than frames with the same semantic content. While methods like SEINE or DynamiCrafter may appear to have lower visual quality than this work, the APM module proposed in this paper, while enhancing content continuity, also restricts the range of generated video content. In my opinion, this is a trade-off with drawbacks. The authors could consider adding experiments to demonstrate that even with CAM and APM, the model can still generate content with semantic variation.
3. This paper employs CAM to ensure short-term consistency in the video, a method that significantly increases the parameter count. In contrast, SEINE’s method, as mentioned, only slightly increases parameters. The paper lacks a clear ablation study to compare the two methods and determine which is superior.

**Weaknesses:**

1. The abstract and introduction repeatedly emphasize that the Appearance Preservation Module (APM)  ensures the natural continuity of object characteristics in generated videos. However, the paper does not  provide metrics similar to CLIP-I to quantify the preservation of subject consistency.
2. When considering long video generation, users typically seek dynamic visuals rather than frames with the  same semantic content. While methods like SEINE or DynamiCrafter may appear to have lower visual  quality than this work, the APM module proposed in this paper, while enhancing content continuity, also  restricts the range of generated video content. In my opinion, this is a trade-off with drawbacks. The  authors could consider adding experiments to demonstrate that even with CAM and APM, the model can  still generate content with semantic variation.
3. This paper employs CAM to ensure short-term consistency in the video, a method that significantly  increases the parameter count. In contrast, SEINE’s method, as mentioned, only slightly increases  parameters. The paper lacks a clear ablation study to compare the two methods and determine which is  superior.

**Questions:**

1. "Table 6" seems incorrectly labeled and should be "Table 1." As far as I can see, there is only one table in the entire paper.
2. In Table. 6, the right side of the table extends beyond the text area, making the layout appear cluttered.

---

### Note · Authors · 2024-11-18

I have read and agree with the venue's withdrawal policy on behalf of myself and my co-authors.